# The Wetland Intrinsic Potential tool: Mapping wetland intrinsic potential through machine learning of multi-scale remote sensing proxies of wetland indicators

Meghan Halabisky [1], Dan Miller [2], Anthony J. Stewart [1], Amy Yahnke[3], Daniel Lorigan [2], Tate Brasel [2] and L. Monika Moskal [1]

[1] School of Environmental and Forest Sciences, University of Washington, Seattle, WA, USA;

[2] Terrainworks, Seattle, WA, USA;

[3] Washington Department of Ecology, Lacey, WA, USA

*Correspondence to*: Meghan Halabisky (halabisk@uw.edu)

**Abstract**

Accurate, un-biased wetland inventories are critical to monitor and protect wetlands from future harm or land conversion. However, most wetland inventories are constructed through manual image interpretation or automated classification of multi-band imagery and are biased towards wetlands that are easy to detect directly in aerial and satellite imagery. Wetlands that are obscured by forest canopy, occur ephemerally, and those without visible standing water are, therefore, often missing from wetland maps. To aid in detection of these cryptic wetlands, we developed the Wetland Intrinsic Potential tool, based on a wetland indicator framework commonly used on the ground to detect wetlands through the presence of hydrophytic vegetation, hydrology, and hydric soils. Our tool uses a random forest model with spatially explicit input variables that represent all three wetland indicators, including novel multi-scale topographic indicators that represent the processes that drive wetland formation, to derive a map of wetland probability. With the ability to include multi-scale topographic indicators that help identify cryptic wetlands, the WIP tool can identify areas conducive to wetland formation while providing a flexible approach that can be adapted to diverse landscapes. For a study area in the Hoh River watershed in Western Washington, USA, classification of the output probability with a threshold of 0.5 provided an overall accuracy of 91.97%. Compared to the National Wetland Inventory, the classified WIP-tool output identified over two times the wetland area and reduced errors of omission from 47.5% to 14.1%, but increased errors of commission from 1.9% to 10.5%. The WIP tool is implemented as an ArcGIs toolbox using a combination of R and python scripts in ArcGIS.

**Key Words: lidar, aerial imagery, soil moisture, forested wetlands**

## 1 Introduction

Wetlands provide a vast array of ecosystem services, including water storage, carbon sequestration, sediment removal, and wildlife habitat (Davidson et al., 2019). Despite their value, over 50% of wetlands worldwide have been lost through draining and filling (Davidson, 2014; Davidson and Finlayson, 2018). Remaining wetlands are surrounded by an increasingly modified landscape that can adversely affect both the condition and function of wetlands (Calhoun et al., 2017; Tiner, 2009). An accurate inventory of wetland locations is necessary to protect wetlands from further land cover changes and degradation. However, in many regions, wetland inventories do not exist or are inaccurate with high errors of omission (Halabisky, 2019). Wetlands under partial or closed canopy, ephemeral wetlands that are flooded for only a portion of the year, and wetlands with no visible standing water (i.e. saturated soils) are often missing from wetland inventories (Halabisky, 2019).

On the ground, wetlands are identified by the presence of three wetland indicators; hydrophytic vegetation, surface hydrology (e.g., inundation or signs of inundation), and hydric soils (Cowardin, 1979). At the landscape scale, however, wetlands are primarily identified using remotely sensed data. Hence, wetland inventories have been commonly created through manual image interpretation by directly identifying wetland characteristics in imagery (e.g., presence of water) or proxies that represent wetland characteristics (e.g., areas of low slope represent areas more likely to be flooded) (Brinson, 1993; Tiner, 1990). In the last decade, there have been great strides in mapping wetlands through automated or semi-automated processes using remotely sensed multispectral data that provide indicators of hydric soil and hydrophytic vegetation (Dronova, 2015; Halabisky, 2019; Lang and McCarty, 2009).

However, small, ephemeral wetlands with dense canopy cover are virtually undetectable in aerial imagery (Figure 1). Even in areas without dense canopy, trees and topography can create shadows in the imagery that can resemble flooded wetlands and confuse automated methods based on spectral features alone. Wetlands with fluctuating water levels or wetlands without visible surface water expression may not be easily detected in the imagery due to a mismatch in the image acquisition timing or poor spectral or spatial image resolution. These cryptic, undetected wetlands can comprise a substantial portion of total wetland area in certain landscapes (Creed et al., 2003; Janisch et al., 2011).

With widespread availability of lidar-derived elevation data, topographic information has been increasingly included as an indicator of wetland potential in analysis of remotely sensed data. Coincident with lidar availability, development of machine learning techniques has enabled analysis of large multivariable data sets at both very high spatial resolution (e.g., Ågren et al., 2021; O'neil et al., 2020; Montgomery et al., 2021; Du et al., 2020) and over very large areas (e.g., Zhang et al., 2023; Woznicki et al., 2019). This work seeks to build on those efforts, with a focus on development of new methods to identify difficult-to-detect, cryptic wetlands. We seek to incorporate the same suite of physical indicators used for ground-based wetland mapping, but using remotely sensed data so that these methods can be applied at regional extents. Because those cryptic wetlands are both difficult to detect with optical or multispectral imagery and typically small, though potentially numerous, we rely on high-resolution lidar elevation data to resolve intrinsic topographic controls on water flux. Recognizing that topographic features that affect water fluxes through a landscape span a large range of spatial extents, we include tools developed to measure topographic attributes over multiple length scales.

**1.1 Topographic and hydrologic indices**

Cryptic wetlands can be indirectly identified by mapping the hydrologic processes driving wetland inundation patterns (Lang et al., 2013; Wu and Lane, 2017). Many studies have shown that delineation of terrain attributes indicative of these processes is effective at predicting wetland locations (Lang et al., 2013; Maxwell et al., 2016; O'Neil et al., 2020, 2018), particularly when these attributes are calculated using high-resolution lidar elevation data. The primary attributes explored in the literature include local topographic position, slope gradient and curvature, the topographic wetness index (TWI), and the cartographic depth-to-water (DTW) (Maxwell et al., 2018). These attributes are calculated using Digital Elevation Models (DEMs), which provide point measures of elevation over a regular grid.

Local topographic position provides a measure of vertical position in the landscape and can differentiate between higher and the low-lying terrain where wetlands tend to occur. There are a variety of methods to calculate local topographic position (Newman et al., 2018), all of which involve comparison of the elevation of a DEM grid point to the elevations of all the other grid points within

a neighborhood of specified radius. The variety of methods for local topographic position differ in how these comparisons are made. The center-cell elevation can be compared to the minimum and maximum elevations or to the mean elevation. That elevation difference can then be used directly or normalized by the range of elevations, by the mean, or by the standard deviation. For

mapping wetland potential, measures of local topographic position are used for identifying landforms where water may tend to accumulate (Branton and Robinson, 2020; Riley et al., 2017).

Slope gradient and curvature are related to the direction and rate of surface and shallow subsurface water flow across the terrain. Water tends to drain quickly from steep slopes and less quickly from lower-gradient slopes. Curvature can indicate areas where

flow directions converge and where rates of flow decrease, both of which are associated with zones of increased soil moisture (Fink and Drohan, 2016).

The topographic wetness index is based on a simple conceptual model of shallow subsurface flow  (Beven and Kirkby, 1979). The depth of soil saturation at a point, or at a DEM cell, is determined by the amount of water flowing to that cell, the degree of

convergence or divergence of the topography there, and the effective velocity (the Darcy velocity) of saturated flow through the soil. Under steady-state rainfall, the amount of water is proportional to the area of the flow tube draining to that DEM cell. The effect of topographic convergence is accounted for by dividing that contributing area by the width of a contour line crossed by water flowing through the cell, giving the specific contributing area $A_s$. The flow velocity is proportional to the tangent of the slope $\theta$. With these definitions, saturation depth varies with $A_s/\tan\theta$. The topographic wetness index is defined as TWI = ln($A_s/$

tan$\theta$). TWI, also called the Compound Topographic Index (CTI), is used as a topographic indicator of relative soil moisture (Kopecký et al., 2021).

The cartographic depth to water (DTW) provides an estimated depth from the ground surface to the saturated zone in the soil column (Murphy et al., 2007). DTW calculates the elevation difference between a DEM grid point and a nearby location of water

at the ground surface, such as a river or lake, which are included as inputs in the model. The location of the associated surface-water point is found by repeatedly finding the adjacent DEM cell with the smallest downslope elevation difference, jumping to that point, and repeating that procedure until surface water is encountered; that is, the least-cost path using slope as the measure of cost. Small DTW values can be good indicators of wetland occurrence (White et al., 2012). Height above the nearest drainage (Nobre et al., 2011) offers an alternative method for estimating depth to the saturated zone. This method finds the elevation

difference between a DEM cell and the surface-water point it drains to based on the downslope flow path traced from each DEM cell (Rennó et al., 2008).

These terrain attributes in various combinations have all been used for wetland identification. The degree of success and the attributes of primary importance vary across studies. This variability reflects intrinsic differences across landscapes (Branton and

Robinson, 2020), but also differences in the spatial resolution of the data used (Fink and Drohan, 2016), preconditioning of that data (O'Neil et al., 2018), and the specific topographic attributes examined. Another source of variability are differences in the spatial scale of the terrain attributes examined.

**1.2 Multi-scale indices for complex topographic features**

All of the terrain measures outlined above are dependent on the length scales over which measurements are calculated. For example, the local topographic position will vary depending on the neighborhood radius used (De Reu et al., 2013). A neighborhood spanning 20 meters will differentiate tree-fall pits and mounds (if resolved by the DEM) while a neighborhood radius spanning kilometers will differentiate valley floors and ridge tops. Gradient and curvature measured over 5 meters length might also detect pits and mounds; gradient and curvature measured over 50 meters will miss those pits and mounds, but will

detect a broad swale. With measurement of any topographic attribute, it is important to match the scales of the landforms we wish to detect and of the processes we wish to characterize.

In regions with complex topography, wetlands are found in topographic features that occur at multiple, interconnected scales (Bertassello et al., 2018; Wu and Lane, 2017). These scales and the degree of interconnectedness vary across and within landscapes,

depending on the landforms and hydrologic processes involved with wetland formation. This variability challenges our ability to use topographic attributes as general indicators of wetland potential. Is a 50-meter-wide depression as important as a 300-meter-wide depression, or a 1000-meter-wide depression? Does it matter if that 50-meter depression is inside of a 1000-meter depression? Likewise, does a depression on a valley floor have the same importance as a depression on a ridge top? Do the relevant scales differ across landscapes? To answer such questions, we must examine topographic attributes over multiple spatial scales.


**1.3 Random Forest**

A large range of factors can be considered for wetlands detection: multispectral imagery, multiple interacting topographic attributes over a range of spatial scales, variations in substrate and landuse. Analysis of such large and diverse datasets has benefited from the development of machine-learning algorithms that do not require assumptions about the statistical distribution of input data

(Maxwell et al., 2018). Non-parametric supervised classification approaches to land cover mapping produce more efficient and accurate results than earlier supervised parametric classification methods (e.g. maximum likelihood) primarily because satellite image data values are not normally distributed (Wulder et al., 2019). Random forest modelling is a commonly used non-parametric classification method (Breiman, 2001), which allows for the use of multiple, correlated input variables that are not normally distributed. These methods are increasingly being used for remote detection of wetlands (Halabisky et al., 2018; Kloiber et al.,

2015; Maxwell et al., 2016; O'Neil et al., 2018; Zhang et al., 2023).

**1.4 Research Goal**

Our goal was to develop a methodology to map intrinsic wetland potential using spatially explicit proxies for three wetland indicators: hydrophytic vegetation, hydrology, and hydric soils. A key objective was to test inclusion of novel multi-scale terrain

indices as a proxy for hydrologic features of variable shapes and sizes. We used a wetland-indicator framework to ensure comprehensive inclusion of wetland characteristics in development of a model, reflecting common wetland identification practices used by wetland ecologists. Framing model development using this framework (and developing a tool for model building with this in mind) helped us ground our approach in wetland ecology, enabling us to more easily bring our domain knowledge into a remote sensing solution. We applied and tested this approach in the Hoh River watershed of Northwest Washington State, a particularly

challenging area to map due to its complex topography, tall and structurally complex forests, and the high variability of wetland types, including many ephemeral wetlands under dense canopy. We have incorporated the methods outlined here into a flexible

ArcGIS toolbox called the Wetland Intrinsic Potential (WIP) Tool to provide an end-to-end workflow that enables users to develop proxies of wetland indicators for their area of interest, including a wide range of topographic indices at multiple scales, to evaluate those indicators using a random forest model, and to use that model to create maps of wetland potential.

## 2 Study area and datasets

Here we define wetlands broadly as wet areas that have one of three wetland indicators; hydric vegetation, hydric soils, or signs of inundation for at least two weeks during the growing season. We included both ephemeral and permanent waterbodies such as rivers and streams in our wetland definition. This decision was driven by the National Wetland Inventory, which includes open water features such as lakes and rivers (Cowardin, 1979).

### 2.1 Study area

Data collection and analyses were performed in the middle and lower Hoh River watershed on the Pacific Northwest coast of Washington State, USA (Figure 1). The Hoh River watershed contains a broad valley filled with alluvial and alpine glacial deposits, with steep alpine zones predominately in marine sedimentary rocks. The main river channel is active and unconfined and has formed terraces from previous higher flows. The Hoh River watershed is part of the Olympic temperate rainforest, receiving between 2.8 and 4.3 meters of precipitation a year, based on PRISM 30-year normals (https://prism.oregonstate.edu/normals/). While the majority of the lower watershed has undergone significant impacts from forest harvest, the upper watershed and area along the coast are within the Olympic National Park (ONP), where forest harvest is prohibited. The trees of the old-growth forest in the ONP can be up to 80 m in height (Harmon and Franklin, 1989), while the lower watershed is dominated by plantation forests managed for timber harvest (Pelt, 2001). The wetlands within the Hoh River watershed are diverse, from precipitation-driven peat bogs to riparian wetlands driven by stream flow inputs, as well as upland wetlands driven by surface water flows and groundwater inputs. The National Wetland Inventory identifies 3,084 hectares of wetlands (not including buffered National Hydrography Dataset streamlines), which comprises 4.4% of the study area (69,558 hectares). Many of the wetlands are under completely closed canopy cover; however, trees in areas of high levels of inundation can display stunted growth.

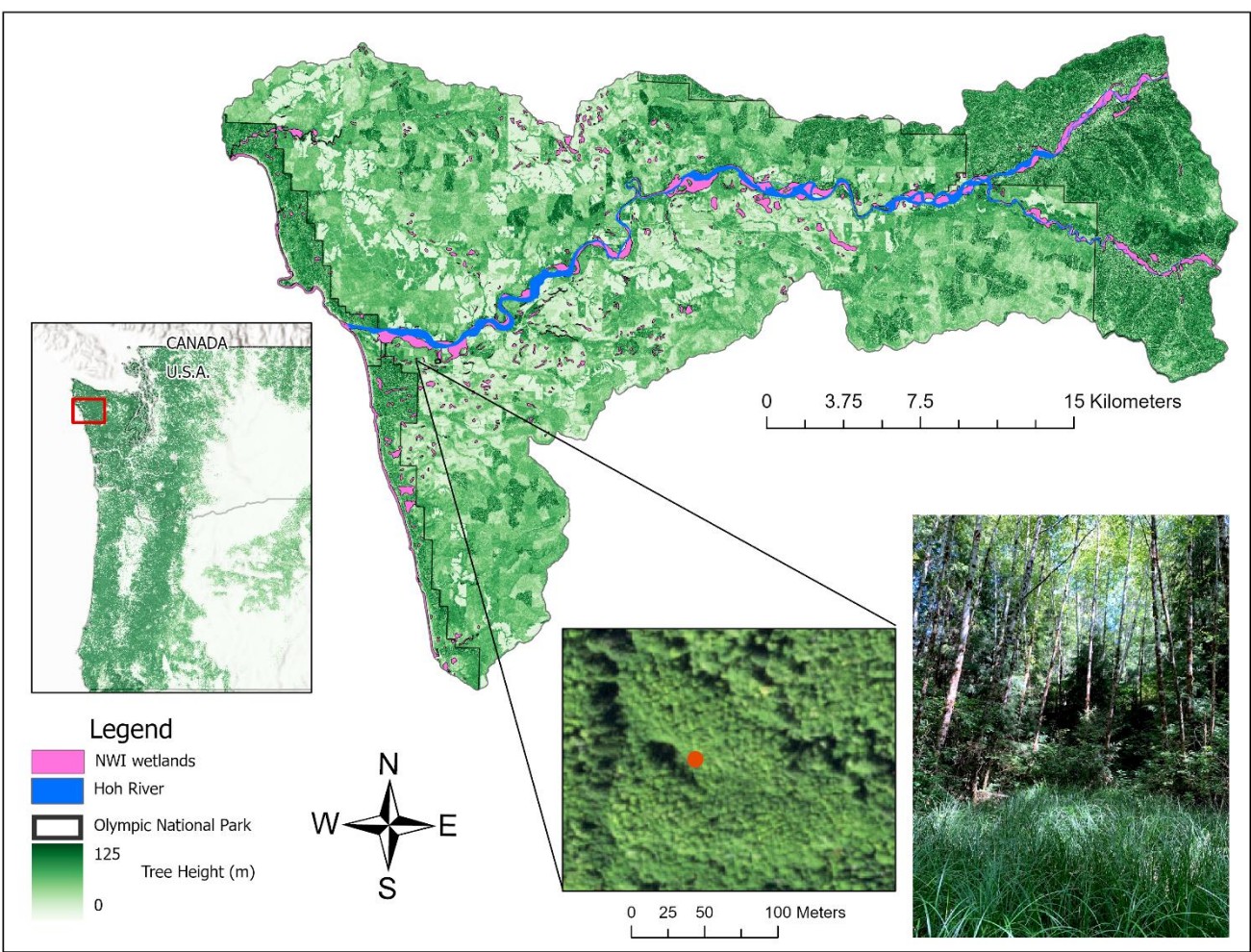

Figure 1. Study area in the Hoh rainforest located in the Pacific Northwest of the United States. Study area shows the variability
in tree height largely determined by a legacy of forestry. National Wetland Inventory (NWI) wetlands are represented as pink
polygons. Areas within Olympic National Park have not been logged. The photo on the right is a picture taken on the ground of
the forested wetland shown in the aerial image from the 2017 National Aerial Imagery Program (NAIP) (orange dot). This wetland
was missed in the NWI and is hard to detect in the aerial imagery. The dark areas in the aerial image are created by shadows from
trees and are not standing water.

**2.2 Data sources**

We used multiple raster and vector datasets as inputs and training data into our random forest model:

1. 4-band aerial imagery acquired by the National Aerial Imagery Program (NAIP) in 2017 at 1m resolution.
2. A DEM and digital surface model (DSM) derived from lidar acquired in 2012 and 2013 by Watershed Sciences at 3-foot pixel resolution and downloaded from the Washington State Department of Natural Resources Lidar data portal (https://lidarportal.dnr.wa.gov/). A DSM is a surface model created from the highest hit object in the lidar point cloud. Subtracting the DEM from the DSM provides estimates of canopy height.
3. Two data layers from the United States Department of Agriculture SSURGO soils data for the Hoh River watershed: the depth to any restricted layer and the hydraulic conductivity.

We used the National Wetland Inventory to create an initial sample training dataset for our preliminary model and for model output comparison. We removed buffered streamlines added into the NWI from the National Hydrography Dataset (NHD) because of the high positional error and the use of a default uniform buffer. Before processing, we re-scaled all of the raster input datasets to match a 4m pixel resolution. The reason for re-scaling to a coarser pixel resolution was to reduce processing time, while still preserving the resolution needed for wetland identification.

## 3 Methods

### 3.1 Developing proxies for wetland indicators

As a first step, we identified spatially explicit proxies that represent wetland indicators for hydrophytic vegetation, hydrology, and hydric soils that we could either derive from aerial imagery, lidar data, or are freely available (i.e., SSURGO soils) (Figure 2). This framework provided us with a systematic way to consider the characteristics used to identify wetlands in the field and in

imagery and determine the ideal proxy that could represent these characteristics as inputs in a random forest model. This framework also allowed us a way to test which group of indicators was most useful in identifying wetlands. We identified datasets that represented proxies based on our own experience and from a thorough literature review.

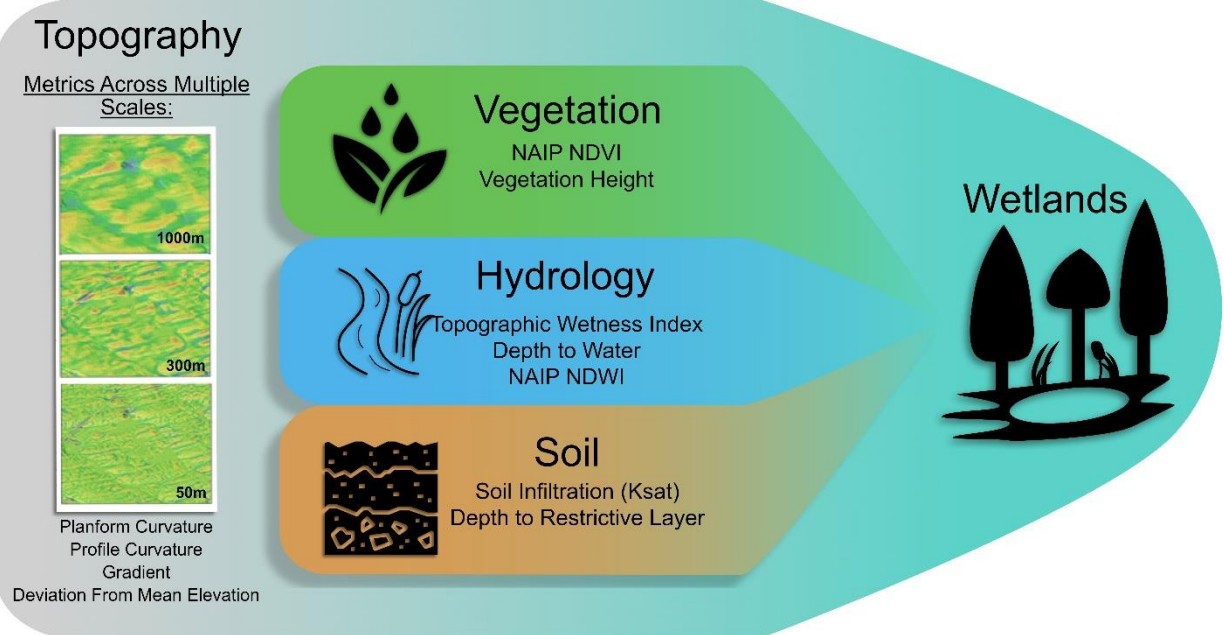

**Figure 2.** Input variables used in the random forest model represent proxies of wetland indicators used for wetland identification.
Topographic indices, calculated at multiple scales, represent areas where water flows and collects. Profile curvature calculated at three different scales; 50m, 300m, & 1000m scale is shown as an example.

### 3.1.1 Hydrologic Indicators

We identified surface water directly in the imagery using the normalized difference water index (NDWI) created from the 2017
NAIP imagery. The NDWI is a normalized band ratio between the near infrared and green bands that is useful to identify open water (McFeeters, 1996). We generated the TWI and the DTW indices using the ArcHydro toolbox in ArcPro using permanent riverine features and waterbodies in the National Hydrography Dataset as input water features for DTW (O'Neil et al., 2018).

In addition to the TWI and the DTW, we explored the use of topographic indices calculated at different length scales.

Gradient and curvature were calculated using the methodology described by Zevenbergen and Thorne, (1987) in which the shape of the ground surface at a DEM grid point is interpolated as a smooth polynomial surface that matches elevations of the grid point and its eight adjacent points. This methodology was modified to use a circular neighbourhood (Shi et al., 2007) of arbitrary radius, with elevations along the circle interpolated from adjacent DEM grid points. This procedure allows estimates of gradient and curvature for each DEM point measured over any length scale, down to the DEM grid size. This is similar to the "local quadratic

regression" described by Newman et al. (2022) but uses a slightly higher-order polynomial with an exact fit to only 9 points, elevation at the current DEM grid point and elevations at 8 equally spaced points on the circumference of a circle of specified radius. This effectively smooths the DEM over the diameter of the circle with no increase in processing time with increasing spatial scale, i.e., with larger circle diameters.

Several topographic position indices have been developed to provide different measures of local relief (Newman et al., 2018)). Of these, deviation from mean elevation (DEV) proved most appropriate for delineating low-lying areas across topographically diverse terrain, $DEV = (z\text{-}z_{mean})/\sigma$, where $z$ is elevation at the point of measurement, $z_{mean}$ is the mean elevation within a neighborhood of specified radius, and $\sigma$ is the standard deviation of elevation within that neighbourhood (Newman et al., 2018). Positive values of DEV indicate the point is higher than the mean of neighboring points (within the specified radius); negative values indicate the

point is lower. Dividing by the standard deviation – a measure of how variable elevations are within the neighborhood – acts to normalize DEV values so that depressions in gentle, low-relief terrain, like broad river valleys, are recognized just as well as depressions in high-relief terrain, like alpine glacial cirques.

We calculated the topographic indices at five different length scales, 50m, 150m, 300m, 500m, 1000m to approximate the

variability of topographic features across the landscape. We visually assessed each topographic index at these scales and decided to only use scales 50m, 300m, and 1000m as they captured the most variability across the landscape and to reduce the number of input datasets to improve processing time.

Topographic indices were calculated using compiled Fortan programs from the Netstream program suite (Miller, 2003). These

programs implement the procedures described above for calculating gradient, curvature, and local relief over any length scale. We developed an ArcGIS Pro toolbox called DEM Utilities for users to create topographic indices at multiple scales.

### 3.1.2 Hydrophytic vegetation and hydric soil indicators

To detect hydric vegetation, we created a normalized difference vegetation index (NDVI) from the 2017 NAIP imagery, rescaled

to 4 meters. NDVI is a normalized band ratio between the near infrared and red bands that is useful at distinguishing wetland from non-wetland vegetation, as well as vegetation that may be stressed from inundation (Halabisky, 2011). We created two raster datasets from the SSURGO soil database, depth to any restricted layer and the hydraulic conductivity, to differentiate the soil properties that influence soil saturation and drainage.

## 3.2 Training Data

Without knowing the location of forested wetlands a priori, it was difficult to develop an efficient and unbiased sampling design. Therefore, to aid in placement of points for a training dataset, we used a stratified random sample from a preliminary wetland model developed from the National Wetland Inventory (NWI, https://www.fws.gov/program/national-wetlands-inventory) for the Hoh River watershed. The preliminary model was based on a random forest model using the topographic indices and trained on 1000 wetland and 2000 non-wetland locations sampled from the NWI. The preliminary model then consisted of a raster of wetland

probability with values from 0 to 1. To generate point locations for training the final model, we randomly placed 600 sample points equally into four strata based on the preliminary wetland probability raster: $0 - 0.25, 0.25 - 0.5, 0.5 - 0.75, 0.75 - 1.0$. This provided an efficient way to identify potential wetland (high probability) and non-wetland (low probability) areas for a balanced point placement, as well as areas where there is high model uncertainty (i.e., probability near 0.5). We felt that stratifying the sample points using the preliminary model would reduce potential bias introduced by referencing the NWI better than if we had solely

used the NWI to create our sample stratification.

Each sample point was evaluated by two analysts and labelled as wetland or upland using available datasets, including a hillshade and slope index from the lidar DEM, pre-existing wetland inventories including the NWI, NAIP imagery, and Forest Practices permits issued by the Washington State Department of Natural Resources, which indicate the presence of wetlands in areas where

timber harvest occurs. If a point could not be determined as a wetland or non-wetland in aerial imagery or any other available datasets, it was marked as unknown. The challenge with this approach is that many of the areas with model certainty close to 0.5 are hard to assess using image interpretation. We made several site visits to ensure that assumptions made in manual image interpretation aligned with the ground truth. Ten percent of the points were visited in the field. In 25 cases where the edge of the wetland was difficult to determine, the point was moved to an area clearly inside or outside the wetland. We removed 2 points

because we could not agree on the label. We were unable to identify any wetlands formed by groundwater expression on slopes with no channel formation to include in the training or validation dataset. Therefore, we expected that the model could not predict or validate the presence of these type of slope wetlands.

## 3.3 Random Forest Model

We used the randomForest package in R (Breiman, 2001) with 598 sample points and 200 trees to train random forest models

using 19 wetland indicators (Figure 2). We decided to use the most complete model with all 19 input data layers based on comparison of the out-of-bag error, a bootstrapped validation approach using sub-selections of the training data. The final model provided a raster showing the probability that a wetland will be found at each DEM grid cell (Figure 3). The Gini coefficient provides a measure of the relative importance of each input indicator in the final model (Figure 4). We classified the wetland probability values into a binary classification of upland and wetland classes using a probability threshold of 0.5.

## 3.4 Model Validation

The WIP tool outputs the probability that a pixel is a wetland. It does not return a binary classification of wetland or upland that could be compared directly to a wetland inventory. To directly compare WIP modeled probabilities to the National Wetland Inventory, we classified all pixels with a modeled probability of 0.5 or above as wetland and all others as not a wetland. The choice of 0.5 for the classification threshold simply reflects that, based on this model output, these pixels are more likely within a wetland than not. A lower threshold would increase the area classified as wetland; a higher threshold would reduce the area. We used this

classification to randomly distribute 100 points within the wetland area and 200 points in the area outside the wetland classification

(i.e. upland). We used the same two-person image interpretation process used for the training sample to label the 300 points. We moved 5 points because we could not detect the wetland edge and removed one point because the analysts could not agree on a label. We used this validation dataset to assess the accuracy of the random forest output and to identify errors of omission and commission.

## 4 Results

Our WIP model classification for the Hoh River watershed identified 6,995 hectares of wetlands using a threshold of 0.5, 2.25 times the area of wetlands mapped by the NWI (3,084 hectares). Model results for the Hoh River watershed can be viewed in detail on an online map available at https://uw.maps.arcgis.com/apps/mapviewer/index.html?webmap=46889ad0fda44662a95efe1559d3f32. The areas identified as wetland had an overall accuracy of 91.97%. The wetland error of commission (false positives) was 10.53% and the error of omission (false negatives, missed wetlands) was 14.14%. In contrast, using the same validation points, the current NWI for the Hoh River watershed had an overall accuracy of 83.95%, with an error of commission of 1.89% and an error of omission of 47.47%.

Table 1. Accuracy assessment for WIP model based on 299 reference points (wetland = 99, upland = 200). A total of 275 of the 299 reference points were classified correctly. Wetland commission error was 10.53% and omission error was 14.14%.

|  | | *Reference Data* | | | |
|---|---|---|---|---|---|
|  |  | Wetland | Upland | Total | Commission error |
| *Model* | Wetland | 85 | 10 | 95 | 10.53% |
| *Results* | Upland | 14 | 190 | 204 | 6.86% |
|  | Total | 99 | 200 | | |
|  | Omission error | 14.14% | 5.00% | Overall Accuracy = 275/299 (91.97%) | |

Gradient calculated at a scale of 50m, tree height (derived from lidar), and local elevation with a scale of 300m were identified as the three variables that contributed the most importance to the model as measured by the Gini importance (Figure 4). Amongst categories shown in Figure 2., there were slightly more topographic indices loading strongly as predictors. Less significant metrics included the coarser 1000m length scales of topography indices, with the exception of the 1000m gradient metric. Other lower metrics of importance included the depth to the restrictive layer and the TWI.

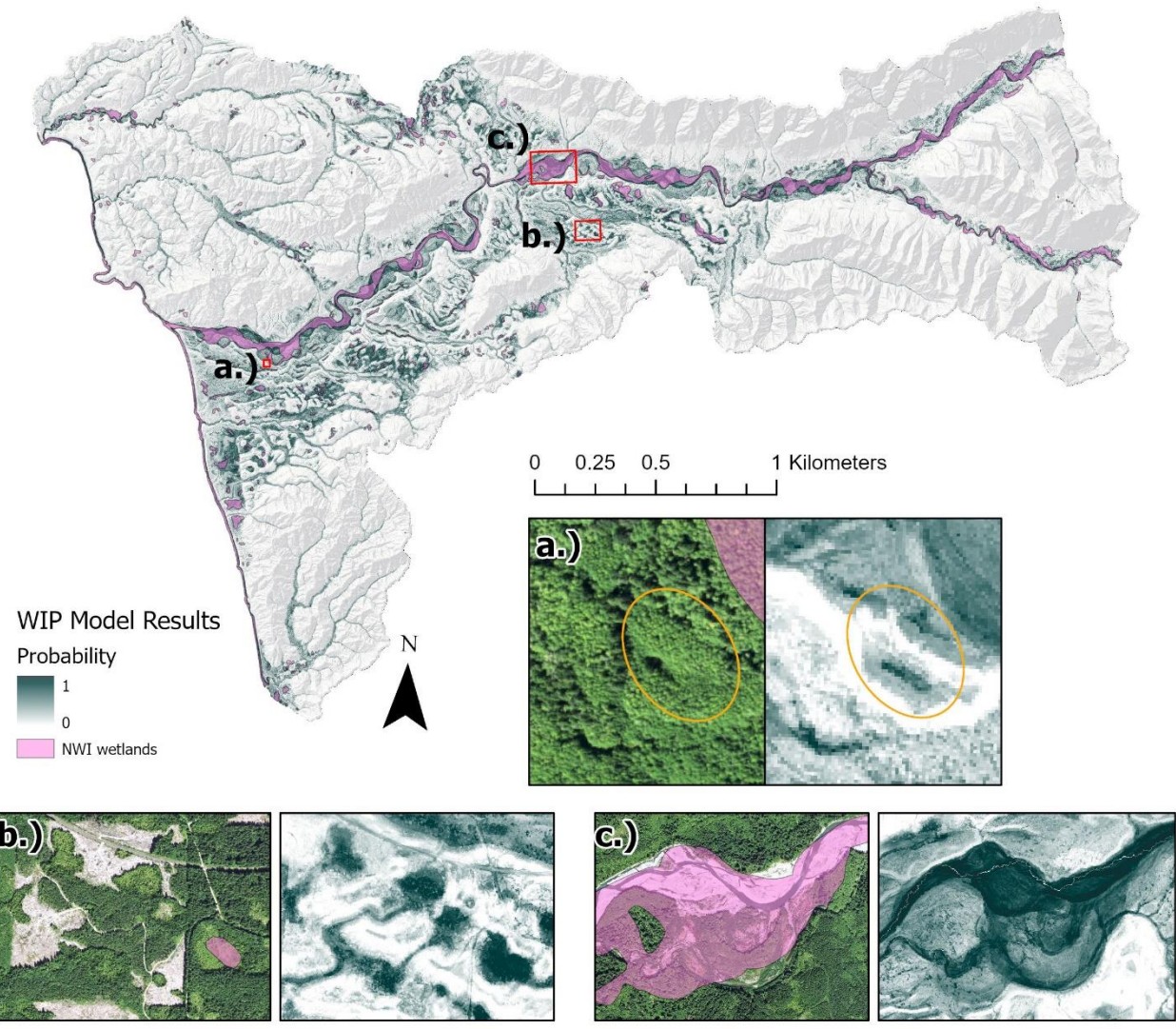

Figure 3. Wetland probability map of the entire study area with three examples: Depressional wetland (a.), peatland (b.), and riverine wetland (c.).

315    Of the 14 labelled wetland points misclassified in the WIP model as upland (errors of omission), 9 of them were within 5 meters of the WIP wetland classification. Conversely, none of the 50 labelled wetland points misclassified as upland (errors of omission) in the NWI model were within 5 meters of the NWI.

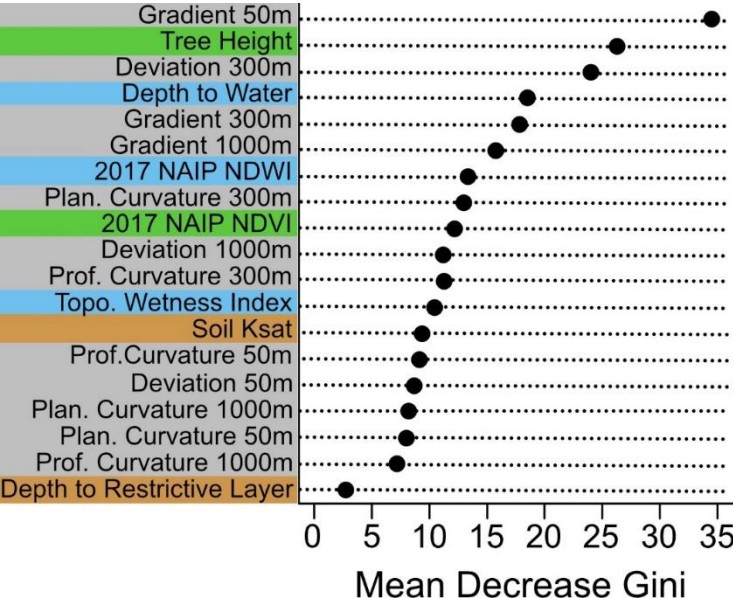

**Figure 4.** Gini coefficient output from the WIP tool random forest model, which is a measure of how each variable contributes to the homogeneity of the nodes and leaves in the resulting random forest. The variables at the top of the chart contributed the most to the model results.

## 5 Discussion

The Wetland Intrinsic Potential tool was designed to improve the detection of wetlands with a specific focus on increasing detection of cryptic wetlands obstructed by vegetation canopy, influenced by shadows from nearby objects and steep topography, and wetlands that do not have visible standing water for some part of the year. Our multi-scale machine learning approach improved the identification of wetlands that were missed in the existing NWI because of these challenging, yet common remote-sensing issues. Using a WIP probability threshold of 0.5 for our validation our model reduced the error of omission by over 33% and the overall accuracy increased by 8% compared to the NWI. The increase in overall accuracy from the NWI was driven by the large reduction in errors of omission. It is important to keep in mind that the NWI has a minimum mapping unit of 0.5 ha, while our WIP tool did not set a minimum mapping and is only limited by the resolution of the input DEM.

For our study area, we found that a combination of proxies representing all three wetland indicators contributed to the overall model importance. However, indicators for hydrologic features and hydrophytic vegetation contributed the most. Specifically, three topographic indices that represent hydrologic features were among the top five input variables. It is unsurprising that measurements of gradient contributed the most, as wetlands are found primarily in areas of low slope. Tree height was the second contributing data layer, which may be driven by both the preference for timber companies to harvest outside of wetlands and the stunted height of trees in wetlands. We did notice that while including tree height improved our model, it also led to an increase in errors of commissions in harvested areas. Users who are interested in identifying wetlands in areas with timber harvest may choose not to include tree height to remove this bias. Measurements of DEV at a scale of 300 was also a top contributing factor, which is useful in identifying medium-sized depressions. Proxies for hydric soils did not contribute as much to the model as other wetland indicator proxies. The hydrologic indicator DTW contributed more than the TWI, however both were lower in importance than seven of the multi-scale terrain indices.

While we used the NWI as a comparison baseline we want to make it explicitly clear that developing a method to replace the NWI
was not our goal here and in no way do we recommend our WIP output as a replacement to the NWI. Rather the WIP output offers
a different paradigm to wetland identification by providing a raster-based product that also provides continuous model probability.
Our WIP probability output in many cases may be preferable to a vector based binary classification for wetland identification
especially for wetlands that do not have clear borders or for use in other landscape models that require continuous raster datasets.
The WIP probability output can also be used to detect wetlands that do not meet the jurisdictional or Cowardin definition of
wetlands, yet still offer substantial ecosystem services such as carbon storage, habitat, and drought refugia. While not a replacement
to the NWI, the WIP tool can be a screening tool to identify omitted wetlands in the NWI (as high as 47.5% in our study area) and
to reduce bias for future NWI updates created through traditional manual photo interpretation.

**5 .1 Model error**

Systematic errors can provide clues for improving model performance. An exploration of the misclassified points shows that, for
this study area, zones with the highest commission error are located around the main river channel. This suggests that large
floodplain and terrace zones should be delineated as categorical variables for input to the random forest model.  Several old river
terraces in old-growth forest stands were also misclassified as wetlands. Further investigation of these points on the ground suggest
that these areas are right on the edge of meeting the definition of a wetland.

Because the majority of the errors of omission were within 5m of a mapped wetland suggests that the model can identify wetland
areas but struggles to accurately delineate the wetland boundary. Some wetlands have clear boundaries, while others have a subtle
wet-dry gradient. In these locations, the edges of wetlands can be hard to delineate on the ground. For wetland types without hard
boundaries, the wetland probability output may provide more realistic information as it picks up the wet-dry gradient.  Object-
based approaches may help identify wetland boundaries in areas with more distinct wetland boundaries, but would require an
additional step of segmentation (Halabisky et al, 2011). Regardless, while the WIP tool can be useful to aid wetland delineations,
standard field techniques on the ground are required for precise wetland delineation. We did not include slope wetlands in our
study because of the difficulty of finding enough samples to train our model for this class of wetlands. We used a threshold of 0.5
and above to classify wetlands for our accuracy assessment.  If users want to lower errors of omission a lower threshold is
recommended. Conversely, if users prefer to avoid over mapping, a higher threshold should be selected.

**5.2 Extension of model to new locations**

The WIP tool is currently available as an ArcGIS toolbox and provides the ability to calculate multi-scale terrain indices. Our
wetland indicator framework allowed us to comprehensively assess a full suite of variables for wetland identification, while
providing a flexible approach that can be adapted to other areas with different topographic features and wetland types. Extension
of the random forest model to new areas requires new training data, which may limit its applicability. The ability of a model to
predict wetland occurrence depends on how well the data used to train the model represent the range of wetland types and locations
that exist on the ground. Our intention was not to develop a model that could be extended to new areas without the collection of
new training data. A model trained on one study area, but run on a different study area, will not produce accurate results if the two
study areas are dissimilar. The importance of different wetland indicators can vary for different study areas, but often the variables
themselves will vary in importance as well. For example, in one watershed that contains many surface-water driven wetlands, the

topographic wetness index may be the most important variable that describes the variability between wetlands and uplands, but in another study area, DTW may be ranked as a more important contributing variable.

For application of the WIP tool in a new area we recommend re-visiting the wetland indicator framework and consider the wetland types in the area of interest and if new remote sensing proxies should be added that we have not considered in our tool. We have found that local knowledge is a critical component of developing solutions that improve model accuracy by identifying data proxies for local conditions. The WIP tool and the wetland indicator framework is designed to be a workflow that can be updated and iteratively improved as new applications and datasets are identified. Indeed, for this project the wetland indicator framework

provided our team a useful framework for testing out existing methods and ultimately led us to identify multi-scale terrain indices that helped identify cryptic forested wetlands and improve our model results. For this study, we tested the WIP tool out in one study area that is considered especially difficult to map. However, the WIP tool has been applied to several new and distinct geographies. Seattle city government used the WIP tool to aid in wetland delineation in the Skagit Basin of Washington (Seattle City Light, 2022); the WIP tool was used to map wetlands in the Island of Hawaii (Tanh et al., 2022), where geology was a key

predictor due to the influence of the volcano.

The WIP tool is designed to be both flexible and allow for iterative improvements from inclusion of additional datasets (e.g. Sentinel-1 data). New datasets can easily be added into the raster stack of input variables in the ArcGIS toolbox. While our goal here was to develop a model with high accuracy and assess multiple wetland indicator proxies, we also realize that our

comprehensive approach may present hurdles to those in areas where some of the data inputs are unavailable. Here we developed a model to optimize for overall accuracy. However, a modified version of the WIP tool with fewer inputs can provide useful results, especially if the probability gradient does not need to be converted into a hard classification. In cases, where a hard wetland classification is not the goal, it may be justifiable to focus only on lidar-derived data inputs as a starting point and include spectral or soils data only if out-of-bag error is not adequate.


In this study area, we used a lidar-derived DEM to create our random forest model input datasets. While lidar data is becoming increasingly more widespread, it is not available everywhere. For areas without lidar coverage, the WIP tool can still be run with a DEM that was not created from a lidar acquisition. We have qualitatively tested out models using a 1/3-arc-second National Elevation Dataset DEM and an IfSAR-derived DEM and found them both to provide potentially adequate results, although at a

coarser spatial resolution.

While we tested this model in a heavily forested area, we believe the WIP tool could be applied to identify wetlands in other landscapes, such as agricultural areas, rangelands, and non-forested areas. However, none of the variables we included in our testing captured water movement influenced by human activity, such as water infrastructure, draining, ditching, or damming.

Therefore, we expect that in areas with high levels of human modification of the hydrology the WIP model may identify areas of intrinsic potential and not necessarily areas that meet current definitions of a wetland.

Finally, our approach was a pixel-based probability and identifies areas of wetland intrinsic potential. However, others may prefer an object-based output (polygons). Object-based segmentation can be run on the WIP tool output to produce polygons and may

improve results for areas where wetlands have more distinct boundaries.

**5.3 Future directions**

While the WIP tool is currently available as an ArcGIS toolbox, we are currently working to integrate components of the WIP into Esri's Wetland Identification Model (WIM), a random forest approach for wetland identification similar to the WIP. Like the WIP the WIM uses elevation-derived wetland indicators for its baseline implementation (O'neil et al., 2020); O'neil et al. (2019); (O'neil et al., 2018), and also accepts other raster-based predictors. WIM is available as part of the Arc Hydro toolset for ArcGIS Pro 2.5 and higher. Specifically, we are working to integrate multi-scale terrain indices and inclusion of point based training data (https://community.esri.com/t5/water-resources-blog/wim-updates-for-arcgis-pro-3/ba-p/1233973. Despite our enthusiasm at integrating the WIP into the WIM, we still see value in a stand-alone open-source tool for those without access to ESRI products. We are currently working with Digital Earth Africa to develop an open-source python based tool to map wetland intrinsic potential using the Open Data Cube (www.digitalearthafrica.org).

**5.4 Model availability**

We designed the WIP tools for this project expecting that they will evolve over time. The scripts and software are licensed as open source and publicly available. The python and R scripts and any new updates for the DEMutilities and Wetland Tool ArcGIS Pro toolboxes are posted to a public github repository at https://github.com/TerrainWorks-Seattle/ForestedWetlands. Bug reports, comments, and feature requests for these toolboxes can be submitted by posting an issue on github. The random forest model can readily accommodate new terrain attributes as explanatory variables and the scripts in the Wetland Tools toolbox can accommodate any input grid that can be imported to ArcGIS. We used the R-ArcGIS Bridge to build the Wetland Tools ArcGIS Pro toolbox that implement scripts that call R functions to build and apply random forest models.

**6 Conclusion**

Wetland inventories are critical sources of data to support wetland conservation prioritization, land use permitting and regulations, monitoring, and wetland research. While wetland features may individually be small, collectively they cover vast areas and contribute to critical ecosystem services. The omission of a large percentage of wetlands within a region impedes our understanding of the total ecosystem services provided by wetlands and how specific land use regulations and policies may impact these services.

Accurate, unbiased wetland inventories are necessary to avoid further degradation and losses of wetlands. The WIP tool was specifically developed to identify cryptic wetlands that are missing from existing wetland inventories, but can also be applied to areas where wetlands have not been mapped well. Our wetland indicator framework, which includes spatial variables representing hydrophytic vegetation, hydrology, and hydric soils, can be used to quantify probability of wetland occurrence, including cryptic wetlands, with high confidence. The inclusion of novel multi-scale topographic attributes greatly improved model results as they were able to capture the variability of topographic features conducive to wetland formation. Our wetland indicator framework provides a flexible approach that can be adapted to identify diverse wetland types across varied landscapes. We expect that the capabilities of the WIP tool will expand over time as users determine the most effective wetland indicators used for identifying wetlands in other regions.

## 7 Code/Data availability

The Fortran programs used to build the raster data sets are licensed under the Gnu Public License[1], version 3. The python and R scripts for the DEMutilities and Wetland Tool ArcGIS Pro toolboxes are posted to a public github repository at https://github.com/TerrainWorks-Seattle/ForestedWetlands. TerrainWorks maintains all software developed during collaborative projects. A comparison between the WIP outputs and the NWI for our study area can be viewed through ArcGIS online map https://uw.maps.arcgis.com/apps/mapviewer/index.html?webmap=46889ad0fda44662a95efe1559d3f32c

## 8 Author contribution

MH, DM, AJS, and LMM designed the sampling, methods, and model design and MH and AJS carried them out. MH, DM, TB, and DL developed the model code and MH performed the simulations. MH prepared the manuscript with contributions from all authors.

## 9 Competing interests

The authors declare that they have no conflict of interest.

## 10 Acknowledgements

The authors would like to acknowledge Vivian Griffey, Sage Ince, Astrid Sanna, Amy Yahnke, and the Washington State CMER Wetland Scientific Advisory group for support collecting training data out in the field. Initial development and testing of the WIP tool was funded by the CMER Wetland Scientific Advisory Group. Final model development was funded through the NASA Carbon Monitoring Science Program (Grant # 80NSSC20K0427).

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
