# Peer review of "The Wetland Intrinsic Potential tool: Mapping wetland intrinsic potential through machine learning of multi-scale remote sensing proxies of wetland indicators"

_EGUsphere, 2022_

## Author Comment (AC1)

[Figure]

Figure 1: Comparison of results for the Hoh watershed to two recently published Global wetland datasets, Xiang et al 2023(a.) and Lane et al. 2023 (b.) to the National Wetland Inventory (c.) and our wetland classification (d.).

Figure 2: Graph showing the relationship between overall accuracy for the WIP model for the Hoh watershed. This graph may be helpful in selection of a probability cutoff (threshold) that divide WIP probability into a binary classification of wetland and upland. Small changes in the cutoff value for the Hoh watershed will impact errors of omission and commission, but won't greatly impact the overall accuracy.

[Figure]

---

## Author Response (AR1)

Review #1

RC1 -1 re: "skeptical about the flexibility of their approach": We appreciate the suggestions and/or questions. We agree that many researchers falsely claim that a method is flexible when it is not and we certainly do not want to overstate the flexibility. To clarify, we believe the WIP tool, not our specific model for the Hoh developed here, is flexible as it can be adapted for new and different geographies. The flexibility of the tool is that the approach is based on a wetland indicator framework realizing that the predictor variables in the random forest model will change based on the wetland types in a selected area of interest. Here, we tested out many predictor variables used in the literature as well as several new multiscale terrain indices. Framing our model around a wetland indicator framework (and building a tool with this in mind) helped us ground the approach in wetland ecology and integrate domain knowledge (wetland ecology) into our remote sensing solution, which we feel could be helpful for others. We set out to build an adaptable approach and tool, instead of a static map or model, which was flexible from the onset and one that can be improved upon over time realizing that users would continue to find new input datasets.

For this study, we tested the WIP tool out in one study area that is considered especially difficult to map. However, the WIP tool has been applied to several new and distinct geographies since. For example, as part of the NASA Develop program the WIP tool was adapted to map wetlands in the Big Island of Hawaii, where geology was a key predictor due to the influence of the volcano (Than et al, 2022). The hope is that as the tool is applied to more areas, new variables can be tested out and identified and can be added as input variables for future applications. The tool in ArcGIS allows for additional input variables to be added. While global and national level inventories will always be important, in many areas like the Washington State local jurisdictions want iterative approaches that can be updated and improved over time. This tool was an attempt to move away from simply producing static maps, but providing tools that can be used and improved through time. We have edited the text to make these points more clear.

RC1-2: We have included the new citations to the manuscript. Thank you. I have compared the Lane et al. 2023, Xiang et al 2023, NWI and the WIP tool (our project) results in this comment. The WIP outperforms all of these datasets substantially in our study area. However, it is perhaps an unfair comparison as the goals of a global dataset are much different than our high resolution watershed specific approach. They are different products and visually the differences are very clear. For these reasons we chose not to include a comparison in the main text of the manuscript. The Lane et al. 2023 dataset aims to improve global wetland maps and uses a much coarser pixel resolution. It does not map wetlands very well across our study area (fig. 1b). As a comment - the Lane et al. 2023 appears to misclassify a large area in the Olympic mountains with few wetlands as a large wetland and seems to pick up riparian area – but not clearly delineating riparian wetlands. Again, this is likely due to the coarse resolution. The Xiang et al 2023 dataset (fig. 1a) misses the majority of the wetlands in the Hoh watershed. The NWI (fig 1c) does an adequate job of mapping wetlands, but misses many of the more difficult to identify wetlands under canopy. All of these datasets seem to illustrate our point that without multiscale terrain metrics it is difficult to identify forested wetlands.

[Figure]

Figure 1: Comparison of results for the Hoh watershed to two recently published Global wetland datasets, Xiang et al 2023(a.) and Lane et al. 2023 (b.) to the National Wetland Inventory (c.) and our wetland classification (d.).

RC1-3 re: "Extension to other case studies." - I agree this approach may not be suitable for areas with no training data. Training data is an essential component of machine learning. However, it is possible to create training data for any area with or without an existing classification. Training data can be created by a simple random sample or stratified random sample, perhaps using a dataset like slope index and then labelling these points as wetland or upland using interpretation of high resolution aerial imagery and topography. In the example in this paper we used an existing wetland dataset (NWI) as training data for a preliminary model for more efficient stratified sampling and labelling. A global wetland dataset, such as the ones mentioned above, could be used to create a preliminary model like the one we developed using the NWI. The purpose of the preliminary model is to reduce the amount of labelling needed to create a robust training dataset. Once a preliminary model is created the user then samples along the model probability, which indicates areas where the model is less certain if a pixel is a wetland or not. This allowed us to collect fewer data samples for training our random forest model – reducing overall effort. Several tools exist to make labelling data more efficient such as Collect Earth Online, Google Earth, and ArcGIS. I have added some text to explain how this can be applied to areas with no wetland inventories. This is being tested out in the Digital Earth Africa platform.

The model has been tested in several watershed across Washington State, British Columbia, Canada, Alaska, Hawaii, and is currently being tested in several watersheds across Africa. However, all of these applications were for management applications and none of these models have yet been published, except the report from the NASA develop team. We have added some text to provide some qualitative information on usability in other areas and cited Than et al, 2022 report for the Big Island of Hawaii. Because a primary focus of this research was to improve errors of omissions, especially forested wetlands, we decided instead of reporting broadly on all of these projects to focus on an intensive validation of our most challenging study area, a densely forested watershed in an old-growth temperate rainforest in the Pacific Northwest where we were able to spend additional time in the field.

RC1-4 re: "selection of a 0.5 threshold"

The goal of this project was not to create a binary classification. We only selected a threshold to create a binary classification to validate the model as a continuous probability estimate cannot be validated. The threshold of 0.5 was used because the model results for that pixel (location on the ground) predicted that it was more likely to be a wetland than not. Because we wanted to test model accuracy we wanted to adhere to using what the model predicted to have a higher likelihood of being a wetland than not. Our goal was not to create a binary classification as we believe probability classification has utility as a standalone product. However, if users want to create a binary classification they can select a threshold to reduce errors of omission or commission. In some cases they may want to minimize false positives and select a higher probability. In other instances, users may be interested in identifying moist forest that does not necessarily meet the criteria of a wetland.

RC1-5 Minor comments:

"Section 2.1. Can you add more specific of the case study? Such as the size of the watershed, the number of wetlands identified by the NWI?"

Thank you for this suggestion. We have added more description to the text. In addition, we have created an ArcGIS online map for readers (and reviewers) to explore the study area, training and validation data, and the pre-existing National Wetland Inventory and our model results. It can be found here.
https://uw.maps.arcgis.com/apps/mapviewer/index.html?webmap=46889ad0fda44662a95efe1559d3f32c

"Figure 1. Can you add the location of the case study also in the map in the inset just above the legend? Either a dot or the boundary of the watershed would be nice to have a sense on where to locate it for readers that are not familiar with the region."

Sorry about that. Somehow the location in the inset map was left off. We have corrected the inset map to include the study area location.

"Table 1. I am not sure I understand what some of the numbers in the table are. Is 85 the number of identified wetlands? The percentage? Please improve the caption of the table so a reader can understand what is going on."

We have improved the caption. 85 is the number of identified wetlands for the validation.

Review #2

RC2-1: "There is a Wetland Identification Model (WIM) that has been available through Arc Hydro since 2020. The WIM methodology is similar to the proposed method in this manuscript, except that WIM only considers DEM data. This manuscript also considers vegetation and soil data. However, these two wetland indicators have also been widely studied in the literature. What's new in the proposed method compared to what has been available in the literature? The O'Neil et. al (2018) has been cited in this manuscript, but the O'Neil et. al 2019 and 2020 papers on the WIM models are not. Why not build upon WIM rather than starting from scratch? "

Building upon the WIM tool is a fantastic suggestion. Development of the WIP model was started before the WIM paper came out and delayed because of COVID. However, since this manuscript was submitted for review (July 2023) we have been put in contact with O'Neil et al (2018, 2019, 2020) and are currently working to integrate the components of the WIP into the WIM tool. This will greatly improve the sustainability of the WIP toolbox within ArcGIS as updates are made by ESRI. We have applied the WIM tool 'as is' to the Hoh watershed and were not able to produce adequate results likely because the curvature metrics used for the WIM do not adequately capture the complex multi-scale terrain associated with wetlands. The most novel part of our method and the finding that has helped us to finally be able to map wetlands in the PNW is our inclusion of multi-scale terrain indices that help identify wetlands of multiple shapes and sizes. We have since shared all our data with ESRI and have met several times to identify ways we can integrate the multi-scale indices into the WIM. In addition we are working with O'Neil to allow for points within the WIM and not just polygons as training data. The updates do far to the WIM have been published on this blog by ESRI https://community.esri.com/t5/water-resources-blog/wim-updates-for-arcgis-pro-3/ba-p/1233973. We will add this update to the manuscript and a link to this blog describing the WIM. We will add the additional O'Neil citations from 2019 and 2020.

To reiterate, the big breakthrough for our research was the inclusion of the multi-scale terrain indices (plan curvature, profile curvature, deviation from local elevation, gradient) as a complement to the other wetland indicator variables. To our knowledge we don't believe that any GIS software can produce these multi-scale terrain indices at this time in the way we have, including the WIM. The inclusion of these multi-scale terrain indices are important complements to other existing datasets like TWI, imagery, soils, etc.. In our watershed these common input datasets did not have as much model importance as our multi-scale terrain indices and may explain why many automated approaches in the PNW fall short. We will emphasize the novelty of our approach more clearly and how it builds on other research more clearly in the discussion.

Despite our enthusiasm at integrating the WIP into the WIM, we still see value in a stand-alone open source tool for those without access to ESRI products. We are currently working with Digital Earth Africa to develop an open source python based tool to map wetland intrinsic potential using the Open Data Cube and have plans to release an R package as soon as time permits.

RC2-2: "Can the authors make the resulting data products (overlaid on NWI layers) available to the public? Maybe through ArcGIS Online and an Earth Engine App so that readers can visually compare the authors' wetland mappings to NWI. Although the commission and omission errors seem reasonable, I am more interested in how the resulting products align with NWI at a fine scale. I am always a bit skeptical about new wetland products unless I can visualize them on an interactive map and compare them with well-known wetland products such as the NWI."

Absolutely. Great idea. All of the training and validation data, and the NWI for comparison are included, as well as the model outputs have been included in the ArcGIS online map for easy review by readers and reviewers. They were also included as supplemental data with this manuscript. We have added this to the text and also mentioned that the datasets are free and open for others to use for model development.

The ArcOnline map is located here:
https://uw.maps.arcgis.com/apps/mapviewer/index.html?webmap=46889ad0fda44662a95efe1559d3f32c

RC2-3: "The proposed wetland tool produced an increased wetland area by 160% compared to NWI. Why? This needs an in-depth discussion. As a reader, I am interested in knowing when the tool works best, and when it fails."

I agree with your suggestion above that providing a visual may be the best way to assess the accuracy. The comparison with the NWI is very different and easy to see in the ArcGIS map online and worth an in-depth comparison. The reason the WIP is able to identify substantially more wetlands is likely due to the inclusion of our novel multi-scale terrain indices that complement the additional input variables. We tested other methods for many years without success, until the discovery and development of these multi-scale terrain indices. The multi-scale terrain indices are helpful because most of the Hoh wetlands are under forest canopy and difficult to identify in imagery alone. Hydrologic indices, such as TWI, which are used in other wetland mapping approaches, are helpful in identifying areas where surface water flows accumulate. The Depth-to-Water is also helpful at identifying wetlands that may have groundwater inputs. However, in the Hoh watershed there are several small swales, depressions, and hummocky areas that are more difficult to identify using hydrologic indices. Additionally, many of the wetlands in the Hoh are precipitation driven peat forming wetlands, and for these the TWI & Depth-to-Water are not particularly helpful. Our novel multi-scale terrain indices are useful at detecting wetlands that are under canopy and occur in nested features of different shapes and sizes (depression, gulch, valley). We tried to set this up in our introduction, but will try and tie the results more directly in the discussion to emphasize the important addition of these multi-scale terrain indices in

our WIP model and explain the model results. Of note, the math is incorrect and the increase is not 160% but rather ~125% - about 2.25 times more wetlands than the NWI. This aligns with other qualitative observations that in the PNW about 50% of the forested wetlands are missing from the NWI.

RC2 – 4: "What is the minimum mapping unit used in this study? Did the authors do any post-processing to reduce the salt-and-pepper effect of the resulting wetland maps? How would that affect the omission and commission error calculations?"

The output of the tool is a probability raster at 4m pixel resolution. The minimum mapping unit, therefore, is the pixel resolution (4 meters). We only selected a threshold to create a binary classification to determine the accuracy of the model in predicting wetlands. However, we feel the usefulness in the tool is the probability raster itself. Depending on the application users can decide to implement further steps, such as OBIA, salt-and-pepper post-processing, manual photo delineation or Cowardin classification, use in the field for sampling, forestry management, or as a model input to predict additional wetland characteristics such as carbon stocks. We did not do any salt-and-pepper post-processing as our primary goal was to assess model accuracy as a probability and any clean-up would distort assessment of the model accuracy.  As a note, this landscape has few depressional wetlands and many complex, hummocky wetlands that occur along a flat terrain. I would imagine if we did implement some post-processing, errors of omission would go up as it would erroneously remove a proportion of these wetlands. Again, this was not the goal of the binary classification so we did not implement any post-processing that would obscure the accuracy assessment of the WIP tool itself

RC2-5: "In terms of the accuracy assessments, did the authors perform both pixel-based and object-based accuracy assessments?"

We did not perform two accuracy assessments. As mentioned above, we did not use an object based approach. The results are a continuous raster of wetland probability. We selected a 0.5 cutoff to test model accuracy. If an object based classification is desired, segmentation may improve results for areas where wetlands have more distinct boundaries. Nevertheless, this tool was not meant to smooth out results, but to be a way to identify all potential wetland area, so we did not implement any post-processing steps such as object-based segmentation or salt and pepper removal. We can add some text to describe how others may improve binary classification if that is their goal through post-processing such as OBIA or filtering.

RC2-6: "The data used in this study are mostly available at the national scale. For example, NAIP and SSURGO data are available at the national scale, and LiDAR data are also available for the majority of the US through the USGS 3DEP program. The training data are derived from NWI, which is also available nationally. I would hope that the proposed tool can be applied to other areas. However, the authors stated in Section 5.2 that their intention was not to develop a model that could be extended to new areas without the collection of new training data. This greatly reduces the transferability of the method and usability of the tool."

Correct. We did not develop a "model" that could be extended to new areas, but rather a method and a tool that could be extended to new areas. Our model was trained on wetland types that occur in the Hoh watershed and therefore the model is inappropriate for other areas, unless of course the watershed is similar to the Hoh (e.g., same ecoregion) or it was used as a preliminary model to help with sampling for further refinement. The downside of machine learning models is that they require a lot of training data. A rule-based approach may be more suitable for areas that have no wetland data at all. However, having created several rule based approaches, it can be difficult identify thresholds without any training data. These thresholds likely are interdependent on other variables and climate making it difficult to implement in complex areas like the Hoh. We will add text and citations for these rule based approaches. However, as mentioned in the response to reviewer #1 a preliminary model can be used to create training data and it is not too difficult to use a preliminary model to develop an efficient sampling scheme to create labelled training points. Several platforms exist making it easier to label data. Our approach is interesting in that we have far fewer training data points than most random forest models and yet we are still able to produce sufficient results. I believe this is because we sampled across the variability of wetlands in an efficient way through the use of a preliminary model classification. A preliminary classification is not needed, sample training points could be created using a random sample or a stratified random sample using a simple layer such as slope index. We will add more text describing methods for areas with low or no training data.

RC2 -7: "Lastly, here are two recently published papers on multi-scale geomorphometric analysis that might be of interest to the authors."

Thank for you these references. We have added them to the manuscript. It is exciting to see the development of these multi-scale geomorphometric indices. Our research has shown these multi-scale indices to be critically important for identifying wetlands in complex forested study areas like the Hoh watershed with variable sized wetland features. We will add some text to the description of how gradient and curvature are calculated:

"Gradient and curvature were calculated using the methodology described by Zevenbergen and Thorne, 200 (1987) in which the shape of the ground surface at a DEM grid point is interpolated as a smooth polynomial surface that matches elevations of the grid point and its eight adjacent points. This methodology was modified to use a circular neighbourhood (Shi et al., 2007) of arbitrary radius, with elevations along the circle interpolated from adjacent DEM grid points. This procedure allows estimates of gradient and curvature for each DEM point measured over any length scale, down to the DEM grid size. This is similar to the "local quadratic regression" described by Newman et al. (2022), but uses a slightly higher-order polynomial with an exact fit to only 9 points, elevation at the current DEM grid point and elevations at 8 equally spaced points on the circumference of a circle of specified radius. This effectively smooths the DEM over the diameter of the circle with no increase in processing time with increasing spatial scale, i.e., with larger circle diameters."

We did not evaluate surface roughness and therefore, did not cite this paper. It is not clear to us how surface roughness (or texture) would be related to topographic controls on groundwater flow, and we were seeking to characterize those topographic controls. Potentially, this could be something worth exploring in further research.

---

## Author Response (AR2)

**Comment 1: "Firstly, the classification results from the WIP Tool appear to be far from perfect, as the overlay on the Esri satellite basemap shows that the wetland area is significantly overestimated, and the commission error seems high. There are many instances of tree shadows and roads being classified as wetlands."**

Thank you for your comment. I humbly disagree that the wetland area is significantly overestimated. However, I can understand your skepticism. Forested wetlands that do not have standing water, but rather saturated soils, are difficult to detect in imagery, especially in evergreen forested areas of the Pacific Northwest US that do not lose their leaves, so it may appear that there is a high error of commission and make it difficult for the reviewer. Even when on the ground these cryptic wetlands can be difficult to find (Figure 1).

Here we provide an example of an area that is correctly mapped by the WIP to highlight the difficulty in identifying wetlands by spectral imagery alone (Figure 2). This example demonstrates how the NWI misses large areas of wetlands – some are easily detected because of the yellow stressed-out vegetation from treed bogs (red arrows), but others are impossible to detect in the imagery alone (yellow arrows). In our region these are often referred to as cryptic wetlands. The areas that appear as tree shadows are in fact small hummocky wetlands. We have included a photo from the field to help demonstrate the challenging nature of this landscape (Figure 1).

To test the assertion that there are many instances of tree shadows and roads being classified as wetlands we re-ran the model without any spectral or tree height data in our input datasets. The results (Figure2) demonstrate that spectral imagery was not an important variable in our model as the model results are similar when spectral imagery was not used (Figure 2, lower right) and agree with our hierarchy of variable importance published in our manuscript. The most important input variables are our unique multi-scale terrain indices created as part of the WIP tool and described in detail in our methods.

[Figure]

**Figure 1:** Field photo of a forested wetland in the PNW. These cryptic wetlands are difficult to detect in aerial imagery. They are statured long enough in the growing season to support wetland vegetation species (skunk cabbage – center), develop hydric soils, and have saturation well into the summer drought months. These

cryptic wetlands provide critical ecosystem services by providing drought refugia, storing large amounts of carbon, and supporting unique species, but they are currently missing from most inventories.

[Figure]

**Figure 1:** Example of a forested wetland area in the Hoh watershed. The image in the upper left shows areas missed by the NWI (pink) pointed out by red and yellow arrows. Red arrows represent wetlands easily detected in the imagery due to the stressed evergreen vegetation. The yellow arrows represent forested wetlands that are difficult to detect in the imagery due to dense canopy but validated on the ground. The image on the upper right shows the output of the WIP probability. It would be difficult to improve upon this map. The WIP picks up all the forested wetlands that are missed. The bottom left shows the binary classification using a threshold of 0.5, which represents a correct estimate of wetlands in the area. The image on the bottom right is the WIP tool re-run without any spectral or vegetation height data used. While removing the spectral imagery reduces the visible sign on roads, we felt that the error of commission from roads was small and decided to keep all input layers in our model as it did improve overall accuracy. I can find no evidence of shadows in either of these outputs. If the shadow effect had been large, it would have been removed once the spectral imagery had been removed from the model. This provides further proof that these wetlands are small wet areas and not caused by tree shadows.

**Comment 2: "Also, the results are quite noisy with numerous tiny and irregular shaped polygons. I think the tool still has a long way to go before it can become a practical dataset complement to NWI."**

We can understand how you may be disappointed if expecting a dataset with a similar look and feel to the NWI. We have added a statement to make it explicitly clear that this was not our goal and in no way do we recommend our WIP output as a replacement to the NWI. Rather the WIP output offers a different paradigm to wetland identification by providing a raster-based product.  Many end users prefer the WIP probability output for wetland identification especially for areas that do not have clear borders as it highlights the gradient they see on the ground and also provides model uncertainty information. Additionally, a raster based gradient of wetland probability can be used for landscape modelling in ways that a vector based dataset cannot, especially useful for Bayesian probability estimates of wetland ecosystem services such as above and below ground carbon stocks (Hudak et al. 2019[1], Moskal et al. 2023[2]) However, there are many users that prefer the look and feel of the NWI and are using this tool as a screening tool in addition to manual photo interpretation to update the NWI, which is still the standard method within the U.S. However, to your point there are many further steps that could be taken to smooth and present a binary classification such as applying a focal smoothing filter. However, applying such a filter may arbitrarily alter the model results in other ways not related to the model inputs. Because presentation and wetland delineation was not the goal of our research, we did not focus on smoothing or clean up. We simply selected a 0.5 threshold to assess accuracy as one cannot assess accuracy for a probability gradient.

We have observed in the field that many of the small tiny and irregular shaped areas are in fact hummocky flats with small depressions that can cover large areas (Figure 2). Because the model output is a pixelated raster model, and not polygons these areas can look irregular and are difficult to delineate through remote sensing imagery or on the ground. However, small wetlands interspersed throughout a landscape can provide critical ecosystem services, even being termed 'wetlandscapes' to reflect their complexity and difficult delineation (Thorslund et al., 2017[3]). However, we believe our continuous wetland probability better reflects potential wetland presence in these landscapes which are currently missing from most inventories.

Above we provide a qualitative example, but our accuracy assessment provides the quantitative analysis to support the strength of the model in this challenging landscape. The error of commission was not substantially high at only 10.24%. That is within the range of other published datasets and if used as a screening tool can be easily dealt with.

We have strengthened our statement in the document to make it clear that we do not intend for the WIP to replace the NWI (Lines 344 – 353). "While we used the NWI as a comparison baseline we want to make it explicitly clear that developing a method to replace the NWI was not our goal here and in no way do we recommend our WIP output as a replacement to the NWI. Rather the WIP output offers a different paradigm to wetland identification by providing a raster-based product that also provides continuous model probability. Our WIP probability output in many cases may be preferable to a vector based binary classification for wetland identification especially for wetlands that do not have clear borders or for use in other landscape models that require continuous raster datasets. The WIP probability output can also be used to detect wetlands that do not meet the jurisdictional or Cowardin
* * *
[1]NASA CMS https://cce-datasharing.gsfc.nasa.gov/cmsprojects/list/h/0/?projType=project&progID=5&projID=4096
[2] NASA CMS https://cce-datasharing.gsfc.nasa.gov/cmsprojects/list/h/0/?projType=project&progID=5&projID=4869).

definition of wetlands, yet still offer substantial ecosystem services such as carbon storage, habitat, and drought refugia. While not a replacement to the NWI, the WIP tool can be a screening tool to identify omitted wetlands in the NWI (as high as 47.5% in our study area) and to reduce bias for future NWI updates created through traditional manual photo interpretation."

To reiterate, while it is fine to convert the WIP tool continuous probability index to a binary classification that was not our goal here but understand that land managers may set thresholds for determining wetland presence to create output products like the NWI. To further address this comment, we have added an additional WIP tool output that provides information to help select a threshold to reduce errors of omission or commission and optimized overall accuracy (Figure 3, left). We used 0.5 to create our binary classification, but if users wanted to remove the commission error they could raise the threshold value without a huge loss in overall accuracy.

[Figure]

**Figure 3:** The left-hand graph shows how the overall accuracy of the WIP model varies across the range of WIP probability values taken from the training reference data. A threshold of 0.50 was chosen in order to validate the results for wetland and upland classification. The right-hand graph shows how the overall accuracy of the classification for the validation data used in the WIP model varies across the range of WIP probability values. Users who are interested in optimizing the overall accuracy may want to reduce the threshold to 0.44.

**Comment 3: "Secondly, it is unclear how the proposed framework would function on other areas, as the authors did not develop a generalized model that could potentially be applied elsewhere. Without addressing the transferability of the method, it would remain a case study."**

The model was initially developed as a tool to identify wetlands that are difficult to detect in the spectral imagery as requested by the Washington State Department of Natural Resources. It was initially tested

on several watersheds across the PNW and published in a report the WA State DNR[3], however the results were not peer-reviewed outside of internal agency peer-review. In order to build statistical confidence in the method we focused analysis on the Hoh watershed, which is considered one of the most difficult areas to map because of the tall evergreen trees and did an intensive effort to create labelled training data and follow up on the ground. This was done to provide confidence in the model and is now being rolled out across the Pacific Northwest, in other parts of the U.S. and now in 3 countries in Africa. The WIP tool was absolutely developed with flexibility in mind with the creation of the wetland indicator framework and draws upon existing literature of proven input datasets for other areas. In some areas other input datasets may be more important model variables.

All machine learning models require training data, and this is true with our model, yet others have shown that machine learning can be a suitable framework and transferred elsewhere when trained or calibrated with local data on new areas (Rußwurm et al 2023). Indeed non-profit organizations exist to make repeatable models available as well as different training datasets so that these models can be trained on new data and calibrated to new locations (https://mlhub.earth/models). We addressed transferability in Section 5.2 (lines 362 – 385). We clearly state that the model itself is not transferable, but the method and the tool can be transferred to other areas and scaled to larger extents with collection of new training data and provide several examples of where it has been successfully run. The method here can be used as an example of how to more efficiently create training data through prelim models run, which can substantially reduce the necessary training data needed to run machine learning model.

Thorslund, Josefin, Jerker Jarsjo, Fernando Jaramillo, James W. Jawitz, Stefano Manzoni, Nandita B. Basu, Sergey R. Chalov, et al. "Wetlands as Large-Scale Nature-Based Solutions: Status and Challenges for Research, Engineering and Management." *Ecological Engineering*, Ecological Engineering of Sustainable Landscapes, 108 (November 1, 2017): 489–97. https://doi.org/10.1016/j.ecoleng.2017.07.012.

Rußwurm, M., Courty, N., Emonet, R., Lefèvre, S., Tuia, D., Tavenard, R. (2023). End-to-end learned early classification of time series for in-season crop type mapping. ISPRS Journal of Photogrammetry and Remote Sensing, 196, 445-456. ISSN 0924-2716. https://doi.org/10.1016/j.isprsjprs.2022.12.016.
* * *
[3] https://www.dnr.wa.gov/publications/bc_fpb_wip_final_report_20210721.pdf